environmental science/climatology/meteorology

*Apis mellifera*, colony winter mortality, weather indicators, apiculture, climatology

**Author for correspondence:**
Benedikt Becsi
e-mail: benedikt.becsi@boku.ac.at

# A biophysical approach to assess weather impacts on honey bee colony winter mortality

Benedikt Becsi[1], Herbert Formayer[1] and Robert Brodschneider[2,3]

[1]Institute of Meteorology and Climatology, and [2]Department of Sustainable Agricultural Systems, Division of Livestock Sciences, University of Natural Resources and Life Sciences, Vienna, Gregor-Mendel-Straße 33, 1180 Vienna, Austria
[3]Institute of Biology, University of Graz, Universitaetsplatz 2/I, 8010 Graz, Austria

BB, 0000-0002-5659-1183; HF, 0000-0002-2126-9696; RB, 0000-0002-2535-0280

The western honey bee (*Apis mellifera*) is one of the most important insects kept by humans, but high colony losses are reported around the world. While the effects of general climatic conditions on colony winter mortality were already demonstrated, no study has investigated specific weather conditions linked to biophysical processes governing colony vitality. Here, we quantify the comparative relevance of four such processes that co-determine the colonies' fitness for wintering during the annual hive management cycle, using a 10-year dataset of winter colony mortality in Austria that includes 266 378 bee colonies. We formulate four process-based hypotheses for wintering success and operationalize them with weather indicators. The empirical data is used to fit simple and multiple linear regression models on different geographical scales. The results show that approximately 20% of winter mortality variability can be explained by the analysed weather conditions, and that it is most sensitive to the duration of extreme cold spells in mid and late winter. Our approach shows the potential of developing weather indicators based on biophysical processes and discusses the way forward for applying them in climate change studies.

## 1. Introduction

The western honey bee (*Apis mellifera*) is an economically important human-kept insect. Next to honey, beeswax, pollen, propolis and royal jelly, honey bees provide a significant ecosystem service: the pollination of wild plants and agricultural crops. For more than a decade, the health status and mortality of managed bee colonies

have been monitored thoroughly in Austria [1–4]. The best-investigated topic is colony mortality during wintering, which also has a significant economic importance [5]. It is surveyed using international standards [6]. The winter months in temperate zones are a critical period for bee colonies: since foraging is not possible, they need to sustain themselves from food stored in the hive. Breeding activity is reduced to a minimum, so worker bees live for months instead of weeks [7]. Wintering is a complex process and is influenced by an abundance of biotic and abiotic factors [8,9]. A field study of winter 2015/2016 in Austria provided evidence for the significance of infestation with the parasitic mite *Varroa destructor* [10]. Other factors that have been proven to impact wintering are beekeeping practices and operation size, queen replacement, *Varroa* control or the availability of melliferous plants [6,11–13]. Related to the latter, the impacts of landscape composition and especially agriculture on honey bee overwintering in Austria were investigated by Kuchling *et al.* [14].

Although colonies can adapt to temperature changes due to thermoregulation [15], the weather has multiple effects on honey bees' foraging and behaviour [16–18], on wintering ability [19] and on *Varroa* control [20]. Until now, there are surprisingly few empirical studies that investigate the correlations between winter mortality and meteorological conditions throughout the year [21–23]. One study quantified these correlations with a large dataset, using 6 years of beekeeper survey data from Austria and monthly averages of temperature, precipitation, global radiation and wind speed [24]. Another study suggested monthly temperature and precipitation averages as co-determinants for the impacts of *Varroa* control methods, using a 5-year dataset of winter colony mortality [21]. While both present an estimation of the sensitivity of winter mortality to annual and monthly average conditions, they do not include any biophysical considerations of the connections between weather and bee colony vitality.

This study aims to close that gap by examining four biophysical processes related to colony fitness for wintering. The processes investigated take place over the course of the whole year: foraging conditions during the flowering period [22,23], cold weather in autumn to signal the start of wintering [8], weather-dependent hive hygiene in winter [25,26] and extreme cold spells as a threat to hive-internal food supply during mid and late winter, the colonies' most precarious time of the year [22,24]. Quantifying and comparing the relevance of these processes for winter survival constitute our main study goals.

We introduce four working hypotheses that define the relations between the biophysical processes and certain weather conditions. Each hypothesis includes an assumption about the direction of the correlation with colony winter mortality.

## 1.1. Forage during the flowering period

To build up strong colonies and sufficient food stores, floral resources in the bees' environment need to be exploited. Favourable weather conditions are required to leave the hive for forage flights. The more days with such favourable weather occur during the flowering period, the more food/energy the colony is able to gather and store and the better prepared it is for the winter [23,27–29]. As honey stores are usually replaced with sugar supplements before wintering, forage conditions affect the spring/summer colony build-up and development, which could affect wintering. We therefore hypothesize the correlation of this biophysical process with winter mortality to be negative.

## 1.2. Cold snaps in autumn trigger wintering

The timing of colonies to start wintering matters for overwintering success. It is assumed that the process of wintering is indirectly triggered i.e. by decreasing temperatures in autumn [30]. If warm periods stretch into late autumn or early winter, sudden cold spells could hit underprepared colonies. Cold spells with adequate duration and intensity during late autumn trigger colony wintering at the right time, reducing the likelihood of colony loss over winter. We assume a negative correlation of cold snaps in autumn with honey bee colony winter mortality.

## 1.3. Hive hygiene in early winter

During the winter months, food stores are consumed by long-lived winter bees. Since bees do not defecate inside the hive to reduce pathogen dispersion, their faeces accumulate in their rectum. Regular snaps of warm weather during the coldest months help bees to leave the hive and defecate. The more regular these mild winter weather conditions occur, the better the hygiene of the hive can be maintained, increasing the bees' vitality [26]. Such snaps of warm weather also facilitate the movement of the winter

cluster to the food stores (see below). We therefore assume a negative correlation of regular warm weather conditions during winter with colony mortality.

## 1.4. Extreme cold spells in winter

Intense and persistent cold spells in mid and late winter can disrupt the food supply inside the hive, as bees require certain minimum temperatures to break the winter cluster and move to the food. At this time of the year, food demand is increased due to the start of egg-laying and brood rearing. High frequency of occurrence and long duration of cold spells could cause increased colony loss rates. Here, we assume the correlation with honey bee colony winter mortality to be positive.

The four hypotheses are operationalized as four weather indicators (two of which consist of two variables, for a total of six variables) and evaluated with empirical data. We use 10 years of beekeeper survey data on honey bee colony losses and meteorological observations from the Austrian weather service ZAMG [31]. Single and multivariate regression analyses are conducted to investigate the correlations at the country level and for Austrian political districts. Quantifying the comparative relevance of the selected weather conditions for colony winter mortality on a district scale could provide useful assistance in hive management for local beekeepers.

# 2. Material and methods

The following section is structured into two subsections. First, the empirical data used to assess the working hypotheses is presented: a 10-year dataset of honey bee colony wintering in Austria and a gridded observational weather dataset. Second, the methods of data processing and analysis are described. Four weather indicators consisting of six variables are defined to operationalize the hypotheses and detect specific weather conditions in the meteorological dataset. Then the statistical approaches used to prepare the data and evaluate the hypothesis are explained.

## 2.1. Data

### 2.1.1. Survey on honey bee colony losses in winter

Data on winter mortality of honey bee colonies presented in this paper originates from an ongoing survey that started in 2008. All beekeepers in Austria were invited to submit at the end of winter the number of wintered colonies, the number of lost colonies during winter and, among other details, their location (or their main location in case of multiple apiary venues) [4]. Location was recorded as municipality and zip code and transformed to GPS coordinates. All datasets were used, regardless of the number of apiaries per operation. The survey considers loss due to queen problems, mortality (dead colonies, empty hives) and natural disasters. The latter is not included in this analysis [13]. In total, the survey yielded 12 779 responses representing 266 378 colonies over all of Austria in the period of 2011 to 2020.

### 2.1.2. Gridded observational weather data

Weather data was obtained from a gridded observational dataset developed by the Austrian weather service ZAMG. The dataset SPARTACUS provides meteorological data on a $1 \times 1$ km grid, interpolated from about 150 homogenized station data series. The data features a daily temporal resolution and is available from 1961 onwards [31]. The following variables from the SPARTACUS dataset were used to calculate the weather indicators employed in this study: daily minimum and maximum temperature, daily mean temperature and daily precipitation sums. Temperature variables are measured between 19.00 CET of the previous day and 19.00 CET of the following day; precipitation sums are counted from 07.00 CET until 07.00 CET of the next day.

## 2.2. Methods

### 2.2.1. Mortality rates

In a first step, GPS coordinates of the respondents' locations are geolocated as point data in QGIS [32]. The points are then aggregated to polygons of the 94 political districts in Austria. This scale is an intermediate between NUTS3 level [33] and municipal level. A more detailed spatial attribution of

apiaries is not realized because the survey only allows the specification of one location per respondent, but apiaries are sometimes distributed over multiple municipalities. Also, aggregating to district level rather than working directly with survey results on point level reduces inaccuracies in individual survey responses. Choosing the polygons based on political boundaries presents an example of the modifiable areal unit problem (MAUP) [34], which describes statistical bias stemming from the arbitrarily chosen unit of spatial aggregation. Here, this bias is accepted as a trade-off for the relevance of our results for public authorities operating within these administrative units.

Bee colony mortality rates are calculated as percentages of lost colonies per district. For each district, an average mortality rate of the survey period (2011–2020) is calculated, including only years with at least five responses (= valid data). In addition, to reduce the leverage of outliers, district averages are only computed if at least 5 years with valid data are available for the respective district. Differences from the average mortality rate were calculated for each year with valid data (i.e. more than five responses). Using absolute percentage points as a difference measure is problematic because of sizeable discrepancies in the distribution of mortality rates between districts. Therefore, mortality anomalies were normalized to standard scores according to Formula 1: Calculation of standard scores for colony mortality rates.

$$Mz_{i,j} = \frac{M_{i,j} - M\mu_i}{M\sigma_i},$$

$Mz_{i,j}$: $z -$ score of mortality in district $i$ and year $j$,

$M_{i,j}$: Mortality rate in district $i$ and year $j$,

$M\mu_i$: Average mortality rate in district $i$

and $\quad M\sigma_i$: Standard deviation of mortality rates in district $i$.

Using normalized standard scores rather than absolute differences increases the comparability of mortality rates between districts. This measure serves as the dependent variable in the statistical analysis described below.

### 2.2.2. Weather indicators

The hypotheses are operationalized in the form of four weather indicators that define the relevant patterns in the meteorological data. Two indicators consist of two variables, bringing the total count of variables to six. In the following, all six variables are referred to as 'indicators' for simplicity.

*Days with optimal flying conditions.* Counted are the days between 1 March and 31 October with daily mean temperatures above 10°C and precipitation sums below 1 mm (dry days). This indicator is similar to the 'flying hours' defined by van Esch *et al.* [22]. It counts the total number of days where bees can forage for food in clear weather conditions during the flying season.

*Cold spells in late autumn.* Counted are periods of at least 7 consecutive days between 15 October and 15 December that start with a day with a minimum temperature below −5°C. Each day of the period has a mean temperature of at most 3°C. If the average daily minimum temperature of the hitherto period exceeds −3°C, it is interrupted. The indicator consists of two variables: total number of days in cold spells and maximum duration of cold spells per year. Such cold spells should occur in the defined time frame to trigger wintering.

*Mild winter weather.* Counted are the number of 9-day moving windows in January and February in which favourable conditions for hygiene flights occur. A window is counted if it contains at least 2 days with a maximum temperature of at least 5°C and precipitation sums of below 1 mm (dry days). The indicator checks for the regular occurrence of such favourable conditions during early and mid-winter. Depending on whether it is a leap year, 51 or 52 such 9-day moving windows are possible in that time frame. The window length was varied between 5 and 10 days in preliminary tests, with 9-day windows producing the best results.

*Cold spells in winter.* Counted are periods of at least 10 consecutive days in January, February and March with a maximum temperature of below 2°C. If the average daily mean temperature of the hitherto period exceeds 0°C the period is interrupted. This indicator also consists of two variables: total number of days in cold spells and maximum duration of cold spells per year. Prolonged cold spells in mid and late winter lead to hive-internal food shortages, resulting in an increase in colony loss rates.

The six indicators feature an annual time resolution (e.g. number of days with optimal flying conditions per year, maximum duration of winter cold spells per year). These annual values were averaged over the period 1981–2010, which constitutes the climatological standard normal [35] until the end of the year 2020. Instead of absolute indicator values, anomalies (absolute differences) from

the climatological period 1981–2010 are used to analyse the relations between weather and bee colony mortality. Using annual absolute values is problematic because some indicators feature a discrete spectrum (e.g. late autumn and winter cold spells). Furthermore, most indicators are temperature dependent and therefore highly correlated with elevation. Using anomalies is a common methodological approach in climatological studies to manage this elevation dependency.

Although apiculture in Austria does extend to areas well above 1000 m.a.s.l., it does not occur frequently at such altitudes. Thus, elevations above 1200 m.a.s.l. were masked out from the gridded weather data and not included in the subsequent analysis. For each district and year, the area median of all masked weather data is calculated. It serves as the independent variable for the statistical analysis described in the next section. Electronic supplementary material, table S.1 provides the correlation structure of the six weather indicators.

## 2.3. Regression analysis

Simple and multiple linear regression with stepwise predictor selection was applied to the whole dataset and to subsets of the data. Other feasible methods include tree-based machine learning algorithms, but for testing and comparing each hypothesis, a less complex and less automated approach was selected. Before the analysis, diagnostic tests were performed to check whether the data satisfies the assumptions of linear regression analysis (normal distribution of the dependent variable, homoscedasticity of residuals). The independent variables are not highly correlated, as shown in the electronic supplementary material, table S.1, except for the two sub-indicators 'cold spells in late autumn' and 'cold spells in winter'. They feature a correlation of 0.97 and 0.94, respectively.

The Austrian-wide analysis examines the relation between mortality rates and indicators for all districts and years, resulting in 700 valid data points (70 districts with valid data × 10 years). The simple linear regression method models the influence of one predictor on the dependent variable (mortality). Naturally, multiple indicators can have combined effects on mortality. Therefore, in addition to simple linear regression, a multivariate linear regression analysis is performed. Due to the high correlation of the two sub-indicator pairs mentioned above, this model includes only one of each sub-indicator, resulting in four weather predictors for the Austrian dataset. Formula 2: Linear regression model for single or multiple predictors, shows the generic equation for single and multiple linear regression models.

$$Y = \alpha + \beta_n * x_n + \varepsilon,$$
$Y$: Predicted value for dependent variable
$\alpha$: Intercept on $y - $ axis
$\beta_n$: Regression coefficients of $n$ independent variables,
$x_n$: Model inputs of $n$ independent variables,
and   $\varepsilon$: Residual error.

To provide relevant results for local beekeepers, both analyses are also performed on the district level. Models are only fitted when at least 5 years of valid mortality data are available, and the weather data show some variability over time ($\sigma > 0$). Regression coefficients, $R^2$-values and $p$-values are calculated for all districts that fulfil the data requirements. The analysis yields each indicator's predictive skill in the district domain.

Multivariate models are fitted with a stepwise algorithm implemented in the statistical software R [36,37]. It fits the model by adding and dropping predictors, starting with a defined set and minimizing the Akaike information criterion (AIC) [38]. Thus, an optimized multiple linear regression model is obtained for each district where the algorithm can determine the AIC. In addition to the comparison of single indicators' relevance, stepwise predictor selection adds combined (additive) effects to the models on district level. A drawback of this downscaling is the small number of data available for each district. Statistical analysis of only 5–10 data points is subject to large uncertainties and overfitting. In the following sections, we will review and quantify this caveat and present reasons for confidence in the results despite small sample sizes.

Electronic supplementary material, table S.2 presents an overview of the statistical analyses carried out in this study, summarizing the scope, parameters and sample size of each model.

# 3. Results

Climatological means were calculated for the six weather indicators and are displayed as maps in figure 1. Regions above 1200 m.a.s.l. are masked out because only lower elevations were considered

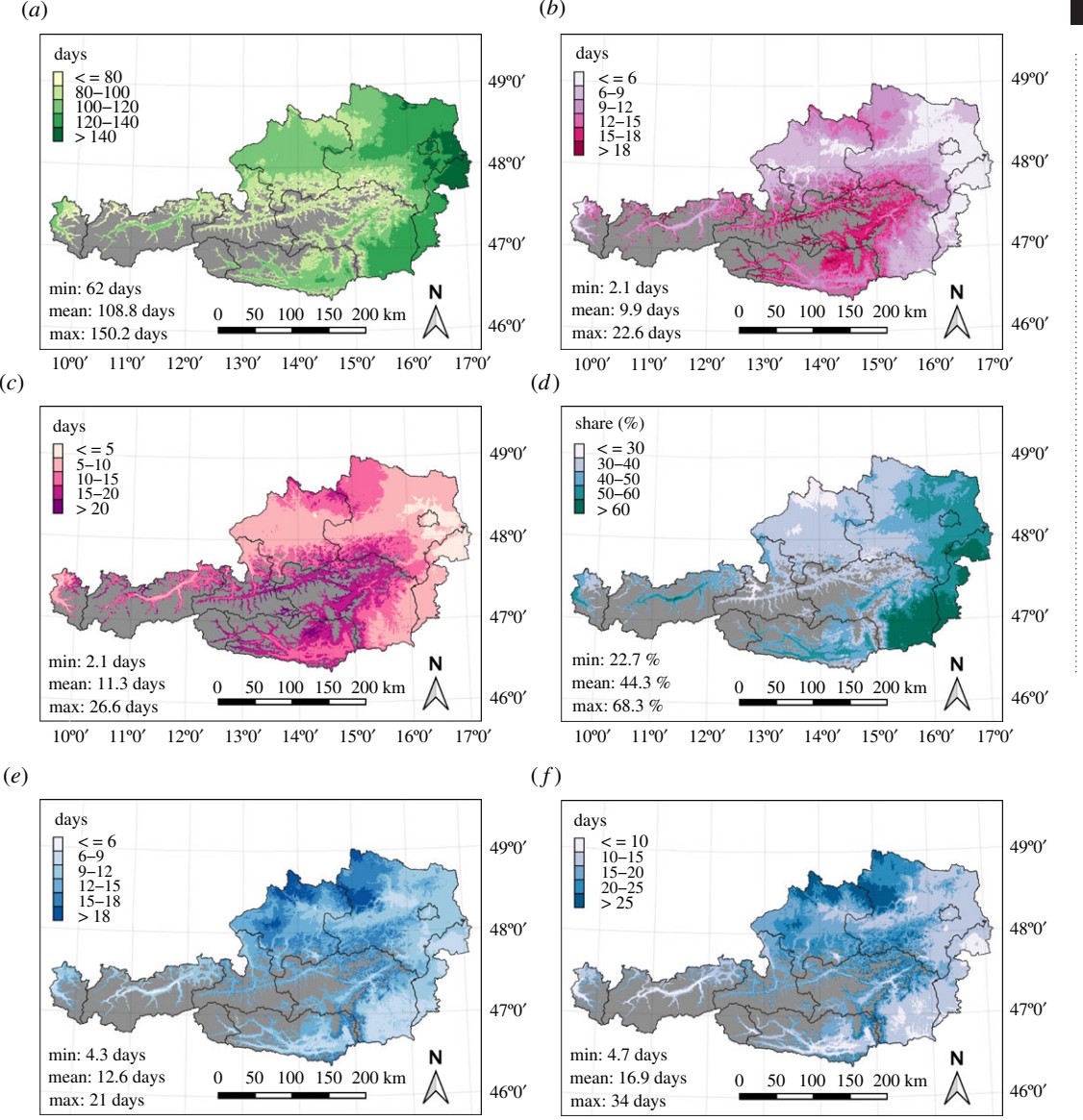

**Figure 1.** Climatological means of the period 1981–2010 for the six selected climate indicators. Regions above 1200 m.a.s.l. are masked out (grey colour). (*a*) Days with optimal flying conditions during the flying season (March–October). (*b*) The maximum duration of cold spells in late autumn (October–December). (*c*) Total number of days in late autumn cold spells (October–December). (*d*) Mild winter weather in January and February (percentage of counted 9-day windows per possible number of 9-day windows). (*e*) The maximum duration of frost waves in winter (January–March). (*f*) Total number of days in frost waves in winter (January–March).

relevant for apiculture in Austria and included in the subsequent analysis. As each indicator is either temperature dependent or temperature and precipitation dependent, elevation is a vital factor for the patterns seen on the maps. While the most favoured regions for the indicator 'days with optimal flying conditions' lie in Vienna, the eastern part of Lower Austria (Marchfeld) and the northern Burgenland ('Seewinkel' region, bordering Lake Neusiedl), the highest share of mild winter weather occurs in the southeastern part of Styria (Grazer Becken) and most regions of Burgenland. Winter cold spell indicators show the coldest regions on the southeastern border of the Alps, and northern Austria ('Waldviertel' in Lower Austria, 'Hausruck' region and northern Upper Austria). For cold spells in late autumn, the eastern pre-Alps and areas north and south of the main Alpine ridge show the highest values. In general, the climatological plots of the six indicators give a good representation of the most advantaged apicultural regions of Austria.

Mean mortality rates were calculated for the study period (2011–2020) and aggregated to Austrian political districts (figure 2*a*). Only years and districts with valid data are shown (see the Methods

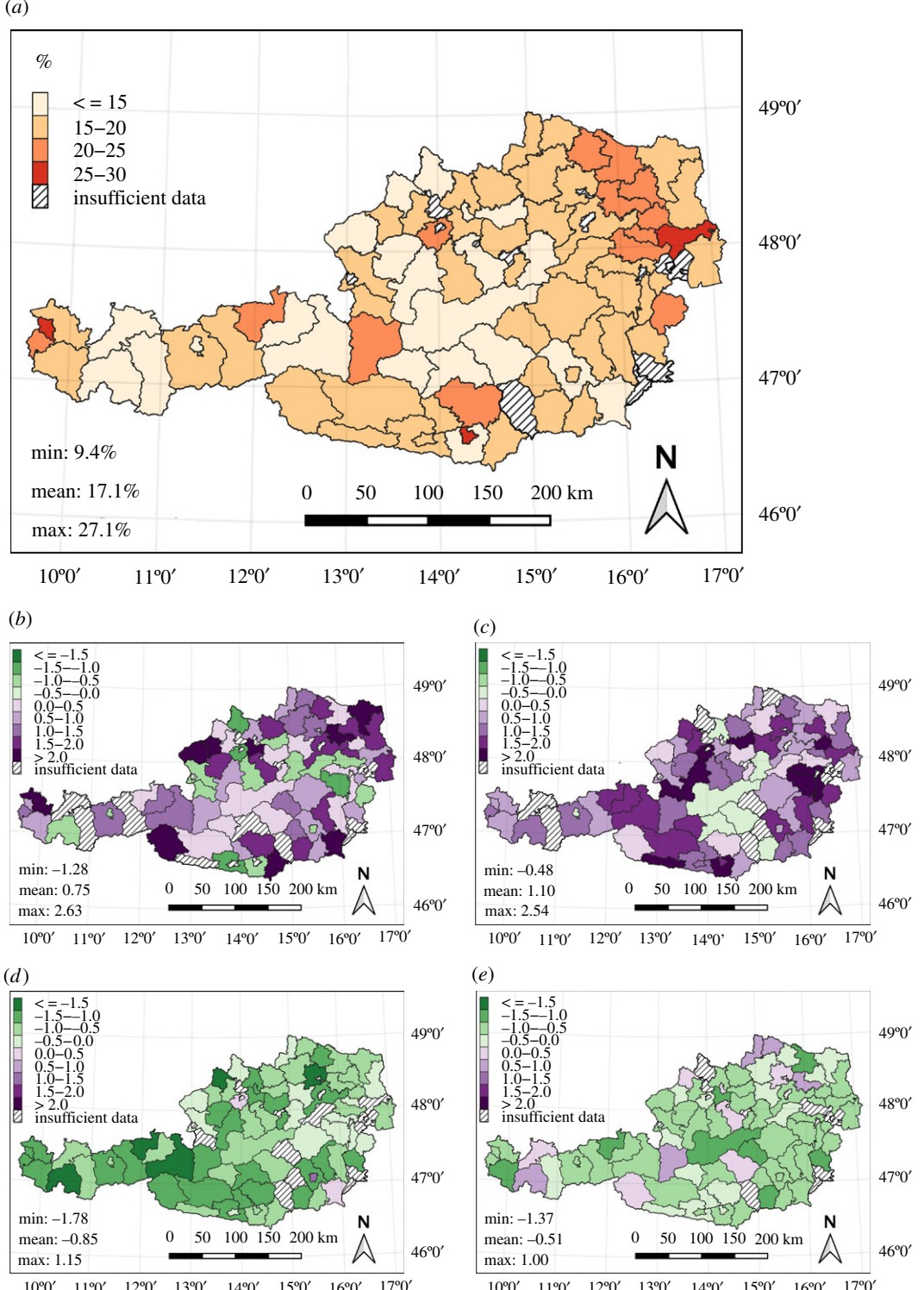

**Figure 2.** Honey bee colony winter mortality rates in Austria. (*a*) Mean mortality rates per district over the study period 2011–2020. Only districts with at least 5 years of valid data are included (compare electronic supplementary material, figure S.2). (*b–e*) Differences of district mortality rates from the mean shown in (*a*), expressed as standard deviations. The selected years are 2012 (*b*), 2015 (*c*), 2016 (*d*) and 2018 (*e*). Green shades mark districts with lower mortality rates than on average, purple colours mark districts with higher mortality rates compared to the mean. Only districts with at least 5 years of valid data are included.

section). The mean mortality rate for wintered bee colonies over the whole of Austria is 17.1%, with a minimum of 9.4% and a maximum of 27.1%. Higher mortality rates generally occur in the eastern part of Austria, especially in the northeast. The central Alpine regions, parts of Upper Austria and

western Tyrol feature lower mortality rates on average. Figure 2b–e shows the difference from the mean mortality rate for each district, displayed as standard deviations for the 4 years 2012, 2015, 2016 and 2018. These years have the highest (2012, 2015) and lowest (2016, 2018) absolute mortality rates within the study period, averaged over Austria. Each of the 4 years shows quite similar standard deviations for most districts.

Some background information about the scope of the survey is provided in the electronic supplementary material, figure S.1 that displays the total number of survey responses over all years for each district, while electronic supplementary material, figure S.2 shows the number of years with valid data per district. Electronic supplementary material, figure S.3 presents the correlation coefficients between the mortality rate time series of each district and the country-wide mortality rate time series. With an average correlation coefficient of 65% and only three districts that are negatively correlated, it reveals a quite homogeneous development of mortality rates over the 10 survey years. This is consistent with the uniform pattern of mortality standard deviations seen in figure 2b–e.

The results from the Austrian-wide simple linear regression analysis are presented in figure 3. The equation of the trend line as well as the adjusted coefficient of determination of the linear model ($R^2$-value, adjusted to the number of data points used for model fitting) and its significance ($p$-value) are annotated in the figure. All trend lines concur with the respective hypothesis's assumed correlation with bee colony winter mortality. The models for optimal flying conditions, mild winter weather and cold spells in late autumn (both maximum duration and number of days in cold spells) all feature negative regression coefficients (decreasing trend). The models for maximum duration and number of days in winter cold spells show positive coefficients (increasing trend). All country-wide models are statistically significant ($p$-value < 0.05) except for the indicator 'number of days in late autumn cold spells'. $R^2$-values range between 0.00 for 'number of days in late autumn cold spells' and 0.06 for 'maximum duration of winter cold spells'. The other indicators, ranked after decreasing $R^2$-values, are 'number of days in winter cold spells' (0.05), 'mild winter weather' (0.03), 'optimal flying conditions' (0.02) and 'maximum duration of cold spells in late autumn' (0.01).

Combined effects of the six weather indicators on colony winter mortality were analysed with multiple linear regression analysis. A multivariate model including four indicators (only the 'maximum duration' sub-indicators for late autumn and winter cold spells were selected) was fitted for all valid data points in Austria. The combined model has an adjusted $R^2$-value of 0.09, meaning it can explain approximately 10% of the country-wide variance in mortality rates. The model is statistically highly significant ($p$-value of $1.865 \times 10^{-14}$) with 695 degrees of freedom ($n = 700$).

The results of the district-level analysis are shown in figure 4. Each panel displays the regression results for one of the six weather indicators. Colours serve as a two-dimensional scale: their hue indicates an increasing (green) or decreasing (orange) slope of the regression trend line. Positive correlations mean that an increase of indicator (anomaly) values cause an increase in mortality rates, and negative correlations mean that an increase of indicator values cause a decrease in mortality rates. The colours' saturation signifies the statistical significance of the fitted model as expressed by the $p$-value. $p$-values higher than 0.1 are considered not statistically significant and are shown in more desaturated colours. The colours give notice whether the indicators' related hypothesis is confirmed by the districts' data. The $R^2$-values noted in the lower left corner of the panels only include districts that match the assumed correlation of the indicators' underlying hypothesis. The results of the district-level regression analysis are summarized in table 1.

For 60–78% of valid districts, the data matches the expected correlation of the indicators' underlying hypothesis. The district-level analysis reveals geographical patterns of indicator relevance. For example, the two 'cold spells in winter' indicators match assumed correlations in districts in the central Alpine regions, in northeastern Austria and Vorarlberg, while the effects are diminished in more peripheral districts. The other indicators that are assumed to be negatively correlated with winter mortality produce better models in the lowlands and pre-Alpine regions, and worse models in the high Alps of central Austria. To highlight these patterns, figure 5 maps the single indicator with the highest adjusted $R^2$-value per district. Green colour marks districts where the indicator 'optimal flying conditions' showed highest $R^2$-values, dark and light purple signify cold spells in late autumn (maximum duration and number of days), orange colour shows mild winter weather and dark/light blue represents winter cold spell indicators (maximum duration and number of days). In general, extreme cold spells in winter are more relevant in the central Alps, while the effects of warmer climates on winter mortality predominate in the lower lying regions of Austria.

For the district St. Poelten (Land) in Lower Austria, which offers a high average $R^2$-value over all indicators (0.33) and a complete set of survey data, the time series of mortality rates and indicator

**Figure 3.** Scatterplots of indicator (x-axis) versus mortality rate (y-axis) anomalies for all years and districts with valid data. Linear regression trend lines are shown as blue dashed lines; the grey area shows the 95% confidence interval. The model's coefficients, significance (p-value) and $R^2$ values are annotated in the graph. Increasing (decreasing) trend lines signify a positive (negative) correlation between the indicator and mortality rates. (n = 700). (a) Days with optimal flying conditions. (b) The maximum duration of late autumn cold spells. (c) Days in late autumn cold spells. (d) Mild winter weather. (e) The maximum duration of winter cold spells. (f) Days in winter cold spells.

values are plotted in figure 6. Mortality is again shown in standard deviations from the mean of the survey period, and indicator values are normalized between 0 and 1 by dividing each value by the maximum of the series. The plot displays the three indicators with the highest $R^2$-values for St. Poelten (Land). Colour hues for indicator time series are similar to the ones shown in figure 4, meaning that green hues are positively correlated with mortality and red/orange hues are negatively correlated. Figure 6 shows that the indicators behave quite as expected from the hypotheses' assumed correlations, with green lines moving in sync with mortality rates and red lines in the opposite direction. Time-series plots of three other districts with good $R^2$-values and complete data are shown in electronic supplementary material, figure S.4.

Electronic supplementary material, figure S.5 provides some additional analysis of the regression results. The plot shows the slope of each indicators' regression trend line as a function of mean indicator values, averaged over the study period 2011–2020. Each data point represents a district. The

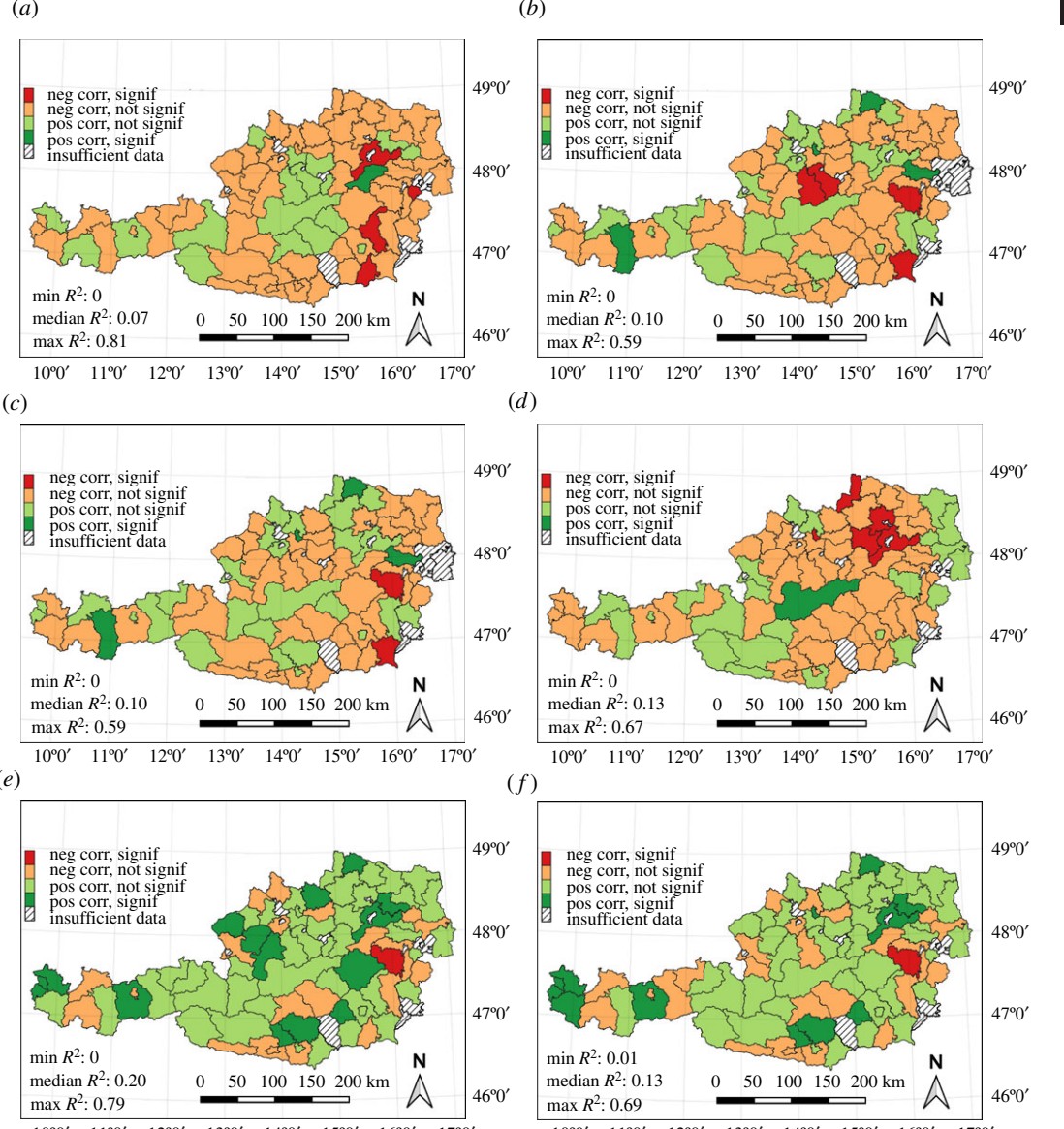

**Figure 4.** Results of simple linear regression analysis for Austrian districts. Red shades show districts with negative regression coefficients; green shades show districts with a positive correlation between indicator and mortality. Darker colours indicate a statistically significant linear model ($p$-value $< = 0.1$). $R^2$ statistics in the lower left of the figures only include districts where the correlation concurs with the respective hypothesis. (a) Days with optimal flying conditions. (b) The maximum duration of late autumn cold spells. (c) Days in late autumn cold spells. (d) Mild winter weather. (e) The maximum duration of winter cold spells. (f) Days in winter cold spells.

number of cases varies slightly (see column 'no. of districts with valid data' in table 1), because in some districts, indicators did not fulfil the data quality requirements. A flat trend line of this function indicates a linear relationship between the indicator and mortality, meaning that mortality would increase/ decrease at the same rate for all Austrian regions. On the other hand, a sloped trend line means an indicator-dependent change of rate of mortality increase or decrease. This effect can be seen for the 'winter cold spells' indicators and is more pronounced for the 'mild winter weather' indicator, albeit in the opposite direction. Mild winter weather seems to have a nonlinear component, meaning that mortality rates decrease more strongly the more windows of mild winter weather occur. These relations can only be interpreted for the value range occurring in Austria. See 'Discussion' section for more details.

Finally, multivariate regression with stepwise predictor selection was performed at the district level. Figure 7 visualizes the statistical significance of the models. Models of districts shown in green are

**Table 1.** Summary of simple linear regression analysis in the country and district domains.

| statistic→ indicator name↓ | assumed correlation from hypothesis | median $R^2$ (district domain)[a][c] | $R^2$ (country domain)[d] | no. of districts with valid data (94 total districts) | no. of districts matching assumed correlation | no. of districts with significant models matching assumed correlation[b] | no. of districts with significant models not matching assumed correlation |
|---|---|---|---|---|---|---|---|
| maximum duration of winter cold spells | positive | 0.20 | 0.06 | 80 | 58 | 15 | 1 |
| days in winter cold spells | positive | 0.13 | 0.05 | 80 | 54 | 12 | 1 |
| mild winter weather | negative | 0.13 | 0.03 | 80 | 56 | 5 | 1 |
| maximum duration of late autumn cold spells | negative | 0.10 | 0.01 | 78 | 50 | 4 | 4 |
| days in late autumn cold spells | negative | 0.10 | 0.00 | 78 | 47 | 2 | 4 |
| days with optimal flying conditions | negative | 0.07 | 0.02 | 80 | 62 | 4 | 1 |

[a]the median is calculated from $R^2$-values of districts with valid data that fit the assumed correlation.
[b]only models that fit the assumed correlation are counted.
[c]see figure 4.
[d]see figure 3.

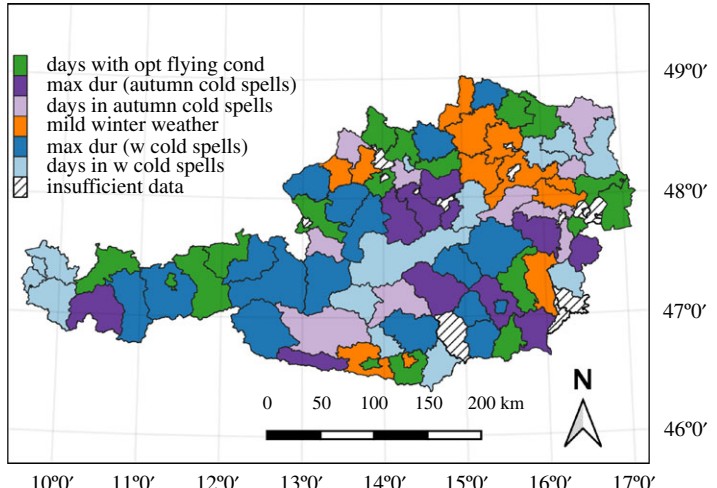

**Figure 5.** Indicator with the highest $R^2$-value for each district. Only models matching the assumptions of the four hypotheses are included.

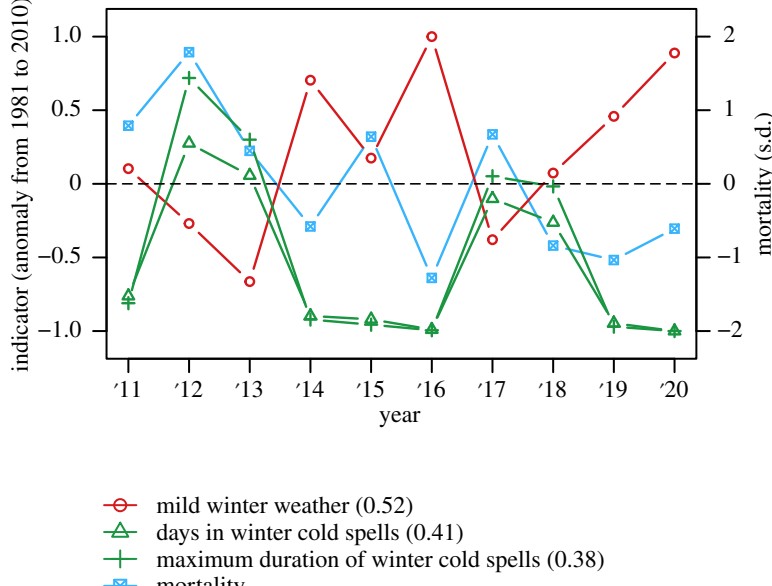

**Figure 6.** Time series of mortality rates (standard deviations) and normalized indicator anomalies for the district Sankt Poelten (Land). Indicator curve colours correspond to the colour code for linear model correlation and significance presented in figure 4. Values in parentheses are $R^2$-values from the district-level simple regression model.

significant, and of districts coloured red are not significant ($p$-value $< 0.1$). Districts coloured grey signify that no predictors had a significant correlation with mortality. The number of districts with valid results decreases to 46 (58%). The median adjusted $R^2$-value of red- and green-coloured districts increases to 0.39. Eleven districts show an $R^2$-value higher than 0.7. Leave-one-out cross-validation resulted in a median $R^2$-value of 0.18 for all districts with valid results, indicating model overfit for the multiple regression models. The regression coefficients consistently match the assumed correlation for most or all the indicators' underlying hypotheses. The full table of model and cross-validation results for each district is provided in the electronic supplementary material, table S3.

## 4. Discussion

Agriculture is undoubtedly one of the most susceptible sectors to climatic changes [39–41]. While agricultural intensification, especially the ever-increasing use of pesticides, is held accountable for the

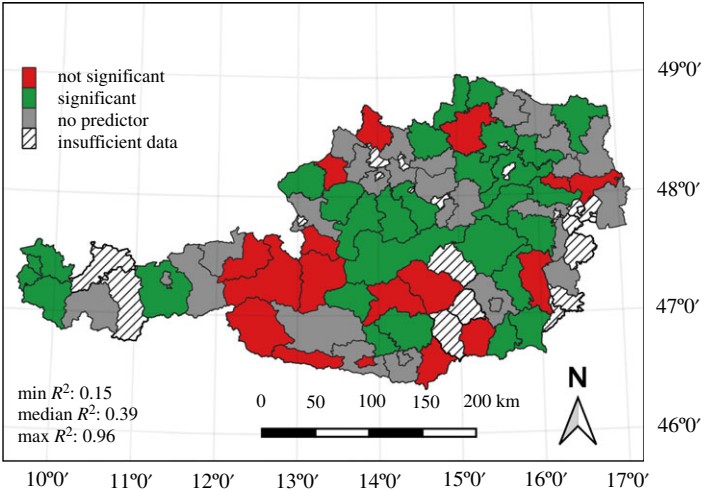

**Figure 7.** Significance of multiple linear regression models fitted with an AIC-based stepwise algorithm on district level. Green colours show statistically significant models ($p < = 0.1$); red colours show statistically not significant models. Grey colour marks districts were models without predictors (only intercept) could be fitted. $R^2$ values in the lower left of the figure include districts with both statistically significant and not significant models (red and green).

worldwide decline of insect populations [42–44], the effects of climate change on insect decline are less clear-cut. Some poikilothermic insects (whose body temperature depends on that of the environment) are assumed to profit from global warming [45]. Biella *et al.* [46] demonstrated that environmental changes (warmer winters) resulted in the natural range expansion of the bumblebee *Bombus haematurus*. For other bumblebee species, limitations in habitat due to climate change have been proven to result in a decrease of abundance or even extinctions [47,48]. Main flight time of four important bumblebee species in Europe extends with higher temperatures [49]. Kammerer *et al.* [50] studied wild-bee abundance in the Mid-Atlantic United States. They found that some wild-bee species show neutral and positive relationships with predicted climate patterns. They further demonstrated that for wild-bee communities in spring and summer/autumn, climatic predictors were more important than landscape.

On the other hand, the influence of weather and climate change on honey bee colonies is often proposed, but poorly studied [51–53]. This paper presents an empirical study on winter colony mortality of honey bees, and the second one for Austria [24]. It proposes four hypotheses about the relations between weather conditions and winter colony loss, based on biophysical processes throughout the year. These hypotheses were evaluated with single and multiple linear regression analysis, using data from a 10-year beekeeper survey and gridded meteorological observations. It is the first study to present weather indicators based on biophysical processes related to colony winter mortality in daily resolution on a $1 \times 1$ km grid.

The average mortality rates of wintered bee colonies per district varied approximately between 10% and 30% over the last 10 years in Austria. High colony losses have previously been explained with two strong drivers, pathogen pressure and hive management [4,10]. A few districts with generally low participation rates in the annual surveys showed particularly high average loss rates. Therefore, a slight bias is suspected for beekeepers to participate in the colony loss survey when they experience unusually high loss rates rather than when they are normal or low. To alleviate the leverage of this 'participation bias' on the mean loss rates, a minimum number of required responses as well as a minimum number of years with sufficient responses per district were introduced in this analysis.

2014/2015 was the winter with the highest overall mortality rate in Austria. This finding is confirmed by Brodschneider *et al.* [2], who report that this winter also entailed the highest colony loss rates in the neighbouring Czech Republic. Often, years with high mortality rates are followed by years with quite low mortality rates because weaker colonies died off, as seen in the consecutive winters 2014/2015 and 2015/2016. The districts generally show very homogeneous standard deviations of mortality rates for most of the surveyed years (electronic supplementary material, figure S.3), hinting at large-scale effects influencing colony mortality. However, this pattern does not seem to extend to larger geographical scales. Gray *et al.* [13] and Brodschneider *et al.* [12,13] present varying mortality rates over Europe for the winter 2017/2018, with Austria being one of few countries with a relatively

homogeneous pattern. The reason for this country-wide consistency of mortality rates can only be speculated at this point.

The assumed correlations of our four hypotheses are confirmed by the empirical data. The Austrian-wide models all concur with the expected trends but show a rather weak quality of fit with $R^2$-values < = 0.06 for the single-predictor models and 0.09 for the multiple regression model. The data reveals a lot of noise (figure 3), since rather short time series of nearly 100 different districts are evaluated together. Moreover, these low $R^2$-values suggest that other factors like agricultural practices, pests and diseases and hive management are missing in the models [9,14,22,54]. Some of those factors are again weather-dependent, making a clear distinction between direct and indirect weather effects difficult [55,56]. In the following paragraphs, the hypotheses' ramifications for winter mortality are discussed in the order of model performance in the country-wide domain, ranked from best to worst.

The assumed positive correlation of 'extreme cold spells in winter' with mortality is most clearly supported by the regression model results. Specifically, the models with an annual maximum duration of cold spells in mid and late winter featured the highest explanatory power. Extreme cold in late winter can lead to an isolation of bees from food stores and thereby to colony loss due to starvation, a common cause for winter loss [1,57]. At this time of the year, food demand is already increasing because of the start of breeding [58]. Our findings join an increasing number of studies that substantiate the negative effect of low temperatures in mid to late winter on colony survival. In Wallonia, Belgium, van Esch *et al.* [22] reported the number of days with frost in February and March as the second most critical factor for successful wintering of honey bees, right after *V. destructor* infestation levels. The first empirical study of colony winter mortality and weather conditions in Austria found that extremely low temperatures in February foremost determined mortality rates [24]. It also noted that an exact temporal attribution of mortality rates with specific meteorological conditions is not possible because the time of colony loss is not recorded in the survey.

The predictor 'mild winter weather' yielded the second best-performing models. The indicator combines information about temperature and precipitation in January and February. While other studies report that high mortality rates correlate with warmer and wetter [21], respectively, dryer [24] monthly average conditions in December and January, their findings are only comparable to a limited extent. The 'mild winter weather' indicator aims at frequently occurring warm and dry conditions on a daily basis and could theoretically produce high values even in relatively cold and wet months. These mild conditions are assumed to have several positive effects on colony vitality, three of which we mention here. First, they allow the bees to leave the hive for hygiene flights. This is an important procedure not only to prevent the spread of diseases [25] but also defecation when food quality is suboptimal. The dates of the first hygiene flights in winter were investigated by Sparks *et al.* [26] who noted a substantial shift to earlier first flight dates during the period 1985–2009 in Poland. No data on first flight dates for Austria has been published yet, but one author's personal observation noted hygiene flights even at the start and middle of winter in some of the last few years. Second, mild temperatures in winter enable the bees to cross from one bee lane (free space between the honeycomb cells) to the next. This is essential so the winter cluster (bees clustering together to a close ball to keep themselves warm) is not cut off from food stores [59]. Finally, it is important for bees to collect water during winter, which they gather at very low ambient temperatures [60].

Foraging conditions during the flowering period were determined by the number of days with optimal flying conditions. The indicator ranked third in model performance. It is quite interesting that weather conditions occurring several months before the start of wintering have an impact on colony mortality [23]. This connection was also found by Switanek *et al.* [24], van Esch *et al.* [22], who defined a similar measure for optimal flying conditions, and Beyer *et al.* [21], who associated cooler and wetter conditions in July with higher mortality rates. And yet, an alleviating effect on colony winter mortality cannot only be directly attributed to the supply foraged by the bees during the flowering period. Usually the food stock is harvested by beekeepers, and if the colony cannot gather sufficient supplies for winter it can be fed [8]. Rather we surmise that many days with optimal foraging conditions lead to colonies experiencing an active summer with a functional and short succession of summer bee generations. This development presumably has a positive effect on colony vitality through autumn and winter. On the other hand, a high number of weather-related disturbances during the foraging period can disrupt breeding cycles, causing worse conditions for colonies even months before wintering [28,29].

The effects of properly timed cold snaps in late autumn to trigger wintering showed the lowest quality of fit in our model results. Reasons for less skilful models might be that delayed wintering

simply did not cause enough noticeable problems during the study period. Moreover, the indicator definition might not be sufficiently accurate to capture the correct triggering events. Although the period length of cold spells was obtained empirically by varying it and selecting the length with the best model results, the temperature thresholds were deterministic. We propose extending the indicator definition to include meteorological conditions before the cold spell event, similar to van Esch *et al.* and Casson *et al.* [22,61]. That way, abrupt temperature changes can be detected more accurately.

The district-level analysis allowed for separate evaluation of models that matched the assumed correlations of our hypotheses. The 'maximum duration of winter cold spells' indicator again showed the highest relevance as a single predictor, with a median $R^2$-value of approximately 0.2 over all districts matching the assumed correlation. In addition, the analysis revealed spatial patterns of the six indicators' compared relevance. Cold spells in winter and late autumn are more relevant in the central Alps, while optimal flying conditions and mild winter weather predominate in the eastern lowlands and some Alpine valleys.

Stepwise predictor selection for multivariate regression models was performed for each district. The data did not meet the stepwise algorithm's criteria in approximately 45% of the districts, but rather high model fits were achieved in the remaining 55%. Compared to the low $R^2$-values of the single-predictor models in the country and district domains, these results show evidence of overfitted models, which is caused by the number of predictors approaching the number of observations. The effect of overfitting was quantified by a leave-one-out cross-validation, resulting in a reduction of approximately 50% of median $R^2$-values for valid districts (see electronic supplementary material, table S.3). In other words, when corrected for overfitting, the multivariate models on the district level exhibit an average quality of fit similar to the single-predictor models including only the 'maximum duration of winter cold spells' indicator. An approach to avoid overfitting would be fitting simpler models (e.g. with a small, fixed set of predictors for all districts), but ultimately the problem stems from the still limited number of available observations.

The regression results of such small samples need to be interpreted with caution, but there are also reasons for confidence in our findings. First, the majority of district-level, single-predictor models fit the assumed correlation of the underlying hypothesis. Second, indicators that performed better on the country-wide scale also feature a higher number of statistically significant models with expected correlations on the district level. Third, when plotting the time series of weather indicators and mortality rates (as we show for selected districts in figure 6 and electronic supplementary material, figure S.4), they quite consistently behave according to the expected correlations. Positively correlated indicators are high when loss rates are high and vice versa. This relation is more pronounced for statistically significant predictors. In summary, these points imply that the single-predictor district models are somewhat trustworthy despite the small number of data points. A similar conclusion regarding data limitations is reached by Calovi *et al.* [23].

The assumption of a linear relationship between indicators and mortality bears some additional note. The correlations analysed in this study are valid for the range of indicator values that occur within Austria and below 1200 m.a.s.l., although they might expand beyond that extent. Thresholds for nonlinear relationships or even reversed correlations surely exist. To name a few examples, increasing temperatures and extended dry periods improve foraging conditions at first, but are bound to have negative impacts on the health and productivity of plant species foraged by bees at some point [62]. In the same manner, warmer winters cannot indefinitely reduce colony mortality. They lead to higher populations of *V. destructor* [10] and in extreme cases, colonies could continue breeding through the winter. If temperature changes induce shifts in growing seasons or activity periods of honey bees, they can result in phenological mismatches [62–64]. Hence the assumed correlations and empirically derived linear relations between weather conditions and honey bee colony mortality should only be interpreted in the context of the analysed dataset. Caution is advised when transferring these correlations to regions with higher elevation, different climate zones or future conditions under climate change.

Another point that, to the best knowledge of the authors, has not been addressed in the literature so far, is how the succession of seasonal weather conditions throughout the year impacts winter mortality. The question is whether beneficial conditions in autumn or winter can compensate for bad flying conditions in summer, and vice versa. With the four periods addressed by our biophysical indicators, there are 16 possible combinations of beneficial (b) or detrimental (d) indicator values over the year (e.g. ddbb, dbdb). An analysis of the mortality response to the different combinations could provide insight into the systematic interdependencies between the indicators. They could possibly amplify (positive feedback loop), attenuate (negative feedback loop) or be independent of each other. Such an

analysis has not been attempted in the scope of this study and would benefit from a longer time series of empirical data to cover as many combinations as possible.

In this study, we analysed the correlations between honey bee colony winter mortality and biophysically based weather indicators. A previous empirical study in Austria presented a model that predicts mortality rates on beekeeper level using monthly or annual averages of temperature, precipitation, global radiation and wind speed [24]. The biophysical approach sharpens the focus of the analysis in two ways: first, we use daily data instead of monthly or annual statistics, which allows us to capture events that might be smoothed out by other conditions in the same month or year. Second, the definitions of the indicators are directly related to biophysical processes relevant for successful wintering. Here, we derived the processes and related thresholds from the literature, but they could also be defined in a co-creation process with experienced beekeepers. Considering the statistical limitations discussed above, the results can help adopt hive management practices to proactively protect colonies against critical weather conditions [8,65]. However, when communicating the results of this study to practitioners, care must be given to not infer causality from the correlations that are analysed here.

Honey bee colony winter mortality is influenced by a multitude of (partly interacting) factors [9,61,66]. Here we only examine the direct and indirect effects of certain weather conditions. For the unexplained variance of winter mortality rates in our models (approx. 80%), data regarding land use [14,22,54], hive management practices, pests and diseases (e.g. *Varroa*) [10,55,67–69], available nutrients [70], phenology [62–64,71], hive management strategies like requeening etc. [4,8,13] need to be taken into account. In addition, bees exhibit an incredible adaptive capacity to different climatic conditions [64], i.e. due to thermoregulation [15,60,72]. Aided by apicultural techniques and optimal site selection they are less vulnerable to climate change than most other insects. Indeed, recent literature discusses honey productivity gain under climate change in Germany [73]. While this may be also the case for more Alpine locations in Austria, the adverse effects of rampant climate change on other important factors, including higher virulence of diseases or the spread of new pests [56,67–69], will dampen the joy of beekeepers over rising temperatures.

# 5. Conclusion

In this paper, we present a novel approach to quantify the effects of weather conditions on honey bee colony winter mortality by defining weather indicators based on biophysical processes. We used data from a standardized beekeeper survey carried out over 10 years in Austria.

Although the regression analysis of the empirical data validated the assumed correlations from our hypotheses, the explanatory power of the models was rather low on different geographical scales. The results showed an explained variance of approximately 10% for the multivariate model in the country-wide domain, and an average of approximately 20% for the best-performing single predictor and the multivariate models on the district scale. The indicator 'maximum duration of extreme cold spells in January, February and March' was determined as the most significant weather-related factor on both geographical scales.

The beekeeper survey data constitutes a rare data treasure that monitors the vitality of honey bee colonies in Austria over a longer period. The biophysical approach could be extended by involving local beekeepers into the definition or prioritization of indicators in transdisciplinary studies. To improve the accuracy of the models, data on *Varroa* infestation and control, diseases, land use, exposure to pesticides and phenological desynchronization could be included as explanatory factors. A systematic evaluation of each factor's individual contribution would supplement the findings of this study and benefit the science on colony winter mortality. Another topic of interest is the interactions of colony winter mortality factors over the seasonal cycle. Furthermore, future research should employ methods to examine nonlinearities and tipping points in the relations between weather conditions and bee colony health to enable the meaningful application of climate model data. This could increase our understanding of the impacts of further climatic change on apiculture.

Data accessibility. The research data supporting this study is available via the Climate Change Centre Austria's Data Centre (https://data.ccca.ac.at/) under an open-access licence (CC-BY), with two exceptions: (a) the raw meteorological dataset SPARTACUS of the Austrian Weather Service (ZAMG), which underlies a commercial licence that prohibits redistribution. Some universities and other institutions have agreements with ZAMG that allow the use of the data for research purposes, which was the case for this study. With the agreement of the journal's Editorial Office, the authors will not be able to make the dataset publicly available on this occasion, but

encourage readers, referees and editors to contact the ZAMG customer service for data access requests: https://www.zamg.ac.at/cms/en/climate/climate. (b) The data regarding the apiary site locations are not publicly available to preserve the privacy of beekeeping operations. In the paper and online repository, only data aggregated to district level is presented. Individual data are, however, available from the authors upon reasonable request. Should you be interested, please contact the corresponding author. The data that is published via the CCCA Data Centre is available under this URL: https://data.ccca.ac.at/group/weather-impacts-on-honey-bee-mortality-in-austria. It consists of four datasets: the raw weather indicators [74], the processed weather indicators [75], the analysis results [76], and the scripts used for computation, statistical analysis and visualization [77]. Detailed descriptions and metadata are provided with the data. For the calculation of indicators, the open-source software NCL (NCAR Command Language) was used [78]. GIS analysis was performed with QGIS [32]. Statistical analysis and visualization was done in R [37].

Authors' contributions. B.B. designed the study, carried out the processing of meteorological data, GIS analysis, statistical analysis and drafted the manuscript; H.F. conceived of the study, participated in the design of the study and critically revised the manuscript; R.B. participated in the design of the study, provided the beekeeper survey field data, participated in data analysis and critically revised the manuscript. All authors gave final approval for publication and agree to be held accountable for the work performed therein.

Competing interests. We declare we have no competing interests.

Funding. This paper builds upon the results of the study 'Impacts of Climate Change on Honey Bee Colonies in Upper Austria' carried out by the authors. It was funded by the section 'Economy' of the Federal State Administration of Upper Austria (grant no. VST 1/781905/7430/001). This paper is part of the doctoral thesis of B.B. Most of his work invested in this paper was unsalaried. The data collection and R.B.'s work were funded by Austrian projects 'Zukunft Biene' (grant no. 100972) and 'Zukunft Biene 2' (grant no. 101295/2).

Acknowledgements. The authors would like to thank the Austrian weather servise (ZAMG) for providing us with the observational weather data used in this study. We are deeply grateful to the two anonymous reviewers who provided excellent feedback and suggestions for improving the manuscript. We also thank Johannes Laimighofer of BOKU Vienna, Institute of Statistics, for his advice and guidance with statistical aspects during the revision of the paper. Open access funding provided by BOKU Vienna Open Access Publishing Fund.

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
