## [Peer Review File · Royal Society Open Science]

Review History

RSOS-210618.R0 (Original submission)

Review form: Reviewer 1

Is the manuscript scientifically sound in its present form?

No

Are the interpretations and conclusions justified by the results?

No

Is the language acceptable?

Yes

Do you have any ethical concerns with this paper?

No

Have you any concerns about statistical analyses in this paper?

No

Recommendation?

Major revision is needed (please make suggestions in comments)

Comments to the Author(s)

Research Article „A biophysical approach to assess weather impacts on honey bee colony winter mortality in Austria” written by Benedikt Becsi, Herbert Formayer, and Robert Brodschneider.

The study focus on colony losses of managed Western honey bee, linking weather conditions throughout the year with Western honey bee colony winter mortality. The approach presented by the Authors is novel and interesting, and the study provide new, important data related to ecology and management of Western honey bee. I congratulate to the Authors for the idea of the study and all the work done. However, in my opinion, the results of the study are misinterpreted and the main, important, finding – that weather conditions, although might contribute to the bee mortality, were among less important factors – remains unseen during discussion of the results, in conclusion, and in an abstract. In my opinion the study, apart from discussion and conclusions, is substantively correct and interesting, therefore I suggest major revision of the current version of the manuscript, mainly related to critical and factual interpretation of the results of the study. Below I provide detailed comments.

Title: why it is important to emphasize Austria in the title? In my opinion the study provides important general knowledge. The current form of the title suggests only local importance.

Page 3, lines 25 and 43 – I suggest to not use phrase “the honey bee” when relating to *Apis mellifera*, since worldwide also other species of honey bees exist. I also suggest to not state in the manuscript that *Apis mellifera* is the most important insect kept by humans without providing appropriate arguments (with citations), rationalizing this statement. Since discussion related to importance of *A. mellifera* is not the aim of this study, the Authors might consider removing this sentence.

Page 4, lines 4-8. Please explain why it is important o emphasize that cited studies were done in Austria and in Luxembourg. Do these studies provide important data and general knowledge about *A. mellifera* ecology and management or they provide only locally important data? If so, why are they important?

Page 4, lines 17-19. Why the main goal of the study was to do the study in Austria? Is it important?

Page 4, lines 44-45. It is oversimplification that bees defecate outside of the hive to avoid the spread of pathogens. Please avoid such statements. Instead, it is better to write that such behaviour may be related to reduction of pathogen dispersion and therefore is evolutionary and ecologically relevant.

Page 5, lines 53-55. I propose to use something like “percentage of colonies lost per district” instead of “ratios of (total number of lost colonies per district /total number of wintered colonies per district) × 100”, since the latter is hard to understand.

Discussion and Conclusion. In my opinion the whole discussion and conclusions should be rewritten, since in the current version of the manuscript the results are misinterpreted. While discussing the results the Authors focus very much on the effects of studied weather conditions on the bee mortality, suggesting that these effects are strong, undoubtful, and that the results of the study prove that the studied weather conditions are important factor driving Western honey bee colony winter mortality. However, the results suggest the opposite. Although the effects of studied weather conditions on the bee mortality were statistically significant, they were also weak. Therefore, the explanatory power of the weather conditions on the bee mortality is low. This means that not studied factors, other than weather conditions, shaped the bee mortality and

that weather conditions are among less important factors. This is important result of the study, and it should be emphasized. I suggest focussing on this result during rewriting of the manuscript.

Page 13, lines 21-25. The Authors suggest here that beekeepers may directly apply the results of this study. I do not agree with this statement, because in my opinion professional scientific knowledge, related among others to modelling, advanced statistics and scientific jargon, is needed for appropriate understanding of this manuscript.

Concluding section of the manuscript is, in my opinion, misleading. It should clearly state that studied factors had low explanatory power.

Review form: Reviewer 2

Is the manuscript scientifically sound in its present form?

Yes

Are the interpretations and conclusions justified by the results?

Yes

Is the language acceptable?

Yes

Do you have any ethical concerns with this paper?

No

Have you any concerns about statistical analyses in this paper?

Yes

Recommendation?

Accept with minor revision (please list in comments)

Comments to the Author(s)

Becsi et al. present an interesting study proposing hypotheses about how climate and weather conditions affect bee hive survival/mortality in apiaries across Austria. The authors are to be commended for clearly formulating and presenting their hypotheses based on mechanistic biophysical processes, and for deriving weather indicators that reflect their hypotheses.

Moreover, the datasets used in this study are extensive in both their spatial extent (across a range of environmental conditions) and sample size. Given the complexity of these data sets and biological processes, the authors have developed an effective strategy for analyzing these data and testing their hypotheses. However, the manuscript would benefit greatly from (i) clarifications of the methods, (ii) explanations for why alternative methods were not used (in specific instances), and (iii) addressing limitations of model performance.

P3: The abstract does not report any quantitative results (e.g., quality of fit of the models). The authors should add information in the abstract which makes it clear that the models do not explain a high amount of the variation in mortality, so that readers who do not review the entire manuscript in detail are aware of the limitations.

P4 (lines 4-8): The authors cite Beyer et al (2018), Van Esch (2020), Calovi et al (2021) later in the manuscript, but should reference these studies here, as examples of empirical studies evaluating the role of meteorological conditions across the year in influencing winter mortality.

P4 (lines 52-53): The authors should add text explaining that the bees must move around to get to honey stores, but cannot break cluster if temps are too low

P4 (line 54): “at this time, food demand is increased due to the start of breeding” maybe rephrase to “start of egg-laying”?

P5 (lines 53-60): The authors justify the mortality z-score because it enables relative comparisons of mortality rates among the districts, and this seems a valid reason. However, why not use binary observations of bee hive mortality and apply a logistic regression (i.e., a generalized linear model) or tree-based machine learning algorithm?

P5 (lines 46-47): The study aggregates the respondents’ point locations to polygons representing the 94 political districts in Austria. This is an example of the modifiable areal unit problem, which can induce statistical bias due to the pattern of spatial aggregation (according to political boundaries, no less!). Have the authors considered how alternative methods of aggregating the points may result in biased observations?

P6 (lines 1-7): So M_z is the z-score of the mortality rate in a district in a given year? Perhaps the equation should include indices and more detailed definitions to more clearly present what is being calculated

P6 (lines 21-51): The authors do not provide information about the correlation structure of dependent weather variables, which would complicate parameter estimation

P6 (line 60): Do the authors mean to say that temperature-dependent indicators are highly correlated with elevation, not sea-level?

P7 (line 5): For some of the districts, these masked areas seem to cover most of the district – is this a problem when evaluating how meteorological conditions influence outcomes in these districts?

P7 (line 14): Again, here would be a good place to mention any strong correlations among the independent variables.

P7 (line 25): The definition of the $x \cdot \beta$ term in Formula 2 should be: β is the regression coefficient/parameter and x are the model inputs.

P7 (line 20-44): It may be simpler to combine Formula 2 and Formula 3 to define the simple linear regression (SLR) and multiple linear regression (MLR) models with one equation. Formula 3 can be re-written with $Y = \alpha + x_n \cdot \beta_n + \epsilon$, where “n” can be one or multiple predictors.

P7 (line 45): Why are the simple LMs necessary, if the effects of each weather indicator on mortality can also be quantified together in an MLR? A clear explanation or table of what models are calibrated to which datasets (e.g., national vs district level, with SLR and MLR applied), the number of parameters in each model, and the sample size of each dataset (including the districts) would be very useful to understand the overall statistical analysis done here.

P20: Table 1 is somewhat unclear. For example in the 1st row: 73% of 80 districts showed expected correlation, but of those only 15 had significant models? Is this interpretation correct? Furthermore, it seems as if only models that match the assumed correlation are included. It may

be helpful to move “No of districts matching assumed correlation” to the column to the right of “No. of districts with valid data”, and then next have a column with “No. of districts with significant models matching assumed correlation” and then a final column with “No. of districts with significant models not matching assumed correlation”.

Table 1, Figure 3, Figure 4: The R^2 values reported here are concerning. Table 1 shows that the quality of fit of the linear models is quite weak ($R^2 < 0.10$) at the country domain, and the R -squared would be expected to decline even further when using the LMs to predict on independent data points. The higher median R^2 values at the district-level are slightly better, but the R^2 values in some districts (e.g., see max R^2 values in Figure 4) are very high and indicative of overfitting (i.e., model complexity / number of parameters approaches the sample size). This makes sense given the reportedly small sample sizes in the districts, but it raises two questions:

- 1) for models with a very low quality of fit, why bother with reporting or further interpretation of coefficients (as in Figure 4)?
- 2) where possible, why not quantify the predictive performance of these models by doing cross-validation to check for overfitting?

The authors stress in their discussion that the data meets the assumptions of the linear model, but do not address that the results show evidence of overfitting. Fitting simpler models with fewer parameters is one option, but with so few data points in some districts our ability to learn from the data is severely restricted. This is a limitation worth noting.

Have the authors analyzed the impact of Varroa mite control on the mortality rates? If so, it would be valuable to include this information to help readers understand how weather versus disease management influence mortality outcomes

Decision letter (RSOS-210618.R0)

Dear Mr Becsi

The Editors assigned to your paper RSOS-210618 "A biophysical approach to assess weather impacts on honey bee colony winter mortality in Austria" have now received comments from reviewers and would like you to revise the paper in accordance with the reviewer comments and any comments from the Editors. Please note this decision does not guarantee eventual acceptance.

Please submit your revised manuscript and required files (see below) no later than 21 days from today's (ie 21-Jun-2021) date. Note: the ScholarOne system will 'lock' if submission of the revision is attempted 21 or more days after the deadline. If you do not think you will be able to meet this deadline please contact the editorial office immediately.

on behalf of Dr Punidan Jeyasingh (Associate Editor) and Pete Smith (Subject Editor)
openscience@royalsociety.org

Associate Editor Comments to Author (Dr Punidan Jeyasingh):

Associate Editor: 1

Comments to the Author:

This manuscript takes a novel approach toward understanding the links between weather and honeybee mortality. The manuscript was assessed by two experts. Both experts were enthusiastic of the work. Nevertheless, they have raised several key issues that require attention. I felt the reviews were fair, clear, and constructive. With much gratitude to the expert reviewers, I invite the authors to incorporate these comments and resubmit a fresh version.

Reviewer comments to Author:

Reviewer: 1

Comments to the Author(s)

Research Article „A biophysical approach to assess weather impacts on honey bee colony winter mortality in Austria” written by Benedikt Becsi, Herbert Formayer, and Robert Brodschneider.

The study focus on colony losses of managed Western honey bee, linking weather conditions throughout the year with Western honey bee colony winter mortality. The approach presented by the Authors is novel and interesting, and the study provide new, important data related to ecology and management of Western honey bee. I congratulate to the Authors for the idea of the study and all the work done. However, in my opinion, the results of the study are misinterpreted and the main, important, finding – that weather conditions, although might contribute to the bee mortality, were among less important factors – remains unseen during discussion of the results, in conclusion, and in an abstract. In my opinion the study, apart from discussion and conclusions, is substantively correct and interesting, therefore I suggest major revision of the current version of the manuscript, mainly related to critical and factual interpretation of the results of the study. Below I provide detailed comments.

Title: why it is important to emphasize Austria in the title? In my opinion the study provides important general knowledge. The current form of the title suggests only local importance.

Page 3, lines 25 and 43 – I suggest to not use phrase “the honey bee” when relating to *Apis mellifera*, since worldwide also other species of honey bees exist. I also suggest to not state in the manuscript that *Apis mellifera* is the most important insect kept by humans without providing appropriate arguments (with citations), rationalizing this statement. Since discussion related to importance of *A. mellifera* is not the aim of this study, the Authors might consider removing this sentence.

Page 4, lines 4-8. Please explain why it is important to emphasize that cited studies were done in Austria and in Luxembourg. Do these studies provide important data and general knowledge about *A. mellifera* ecology and management or they provide only locally important data? If so, why are they important?

Page 4, lines 17-19. Why the main goal of the study was to do the study in Austria? Is it important?

Page 4, lines 44-45. It is oversimplification that bees defecate outside of the hive to avoid the spread of pathogens. Please avoid such statements. Instead, it is better to write that such behaviour may be related to reduction of pathogen dispersion and therefore is evolutionary and ecologically relevant.

Page 5, lines 53-55. I propose to use something like “percentage of colonies lost per district” instead of “ratios of (total number of lost colonies per district / total number of wintered colonies per district) × 100”, since the latter is hard to understand.

Discussion and Conclusion. In my opinion the whole discussion and conclusions should be rewritten, since in the current version of the manuscript the results are misinterpreted. While discussing the results the Authors focus very much on the effects of studied weather conditions on the bee mortality, suggesting that these effects are strong, undoubtful, and that the results of the study prove that the studied weather conditions are important factor driving Western honey bee colony winter mortality. However, the results suggest the opposite. Although the effects of studied weather conditions on the bee mortality were statistically significant, they were also weak. Therefore, the explanatory power of the weather conditions on the bee mortality is low. This means that not studied factors, other than weather conditions, shaped the bee mortality and that weather conditions are among less important factors. This is important result of the study, and it should be emphasized. I suggest focussing on this result during rewriting of the manuscript.

Page 13, lines 21-25. The Authors suggest here that beekeepers may directly apply the results of this study. I do not agree with this statement, because in my opinion professional scientific knowledge, related among others to modelling, advanced statistics and scientific jargon, is needed for appropriate understanding of this manuscript.

Concluding section of the manuscript is, in my opinion, misleading. It should clearly state that studied factors had low explanatory power.

Reviewer: 2

Comments to the Author(s)

Becki et al. present an interesting study proposing hypotheses about how climate and weather conditions affect bee hive survival/mortality in apiaries across Austria. The authors are to be

commended for clearly formulating and presenting their hypotheses based on mechanistic biophysical processes, and for deriving weather indicators that reflect their hypotheses. Moreover, the datasets used in this study are extensive in both their spatial extent (across a range of environmental conditions) and sample size. Given the complexity of these data sets and biological processes, the authors have developed an effective strategy for analyzing these data and testing their hypotheses. However, the manuscript would benefit greatly from (i) clarifications of the methods, (ii) explanations for why alternative methods were not used (in specific instances), and (iii) addressing limitations of model performance.

P3: The abstract does not report any quantitative results (e.g., quality of fit of the models). The authors should add information in the abstract which makes it clear that the models do not explain a high amount of the variation in mortality, so that readers who do not review the entire manuscript in detail are aware of the limitations.

P4 (lines 4-8): The authors cite Beyer et al (2018), Van Esch (2020), Calovi et al (2021) later in the manuscript, but should reference these studies here, as examples of empirical studies evaluating the role of meteorological conditions across the year in influencing winter mortality.

P4 (lines 52-53): The authors should add text explaining that the bees must move around to get to honey stores, but cannot break cluster if temps are too low

P4 (line 54): "at this time, food demand is increased due to the start of breeding" maybe rephrase to "start of egg-laying"?

P5 (lines 53-60): The authors justify the mortality z-score because it enables relative comparisons of mortality rates among the districts, and this seems a valid reason. However, why not use binary observations of bee hive mortality and apply a logistic regression (i.e., a generalized linear model) or tree-based machine learning algorithm?

P5 (lines 46-47): The study aggregates the respondents' point locations to polygons representing the 94 political districts in Austria. This is an example of the modifiable areal unit problem, which can induce statistical bias due to the pattern of spatial aggregation (according to political boundaries, no less!). Have the authors considered how alternative methods of aggregating the points may result in biased observations?

P6 (lines 1-7): So M_z is the z-score of the mortality rate in a district in a given year? Perhaps the equation should include indices and more detailed definitions to more clearly present what is being calculated

P6 (lines 21-51): The authors do not provide information about the correlation structure of dependent weather variables, which would complicate parameter estimation

P6 (line 60): Do the authors mean to say that temperature-dependent indicators are highly correlated with elevation, not sea-level?

P7 (line 5): For some of the districts, these masked areas seem to cover most of the district – is this a problem when evaluating how meteorological conditions influence outcomes in these districts?

P7 (line 14): Again, here would be a good place to mention any strong correlations among the independent variables.

P7 (line 25): The definition of the $x^*\beta$ term in Formula 2 should be: β is the regression coefficient/parameter and x are the model inputs.

P7 (line 20-44): It may be simpler to combine Formula 2 and Formula 3 to define the simple linear regression (SLR) and multiple linear regression (MLR) models with one equation. Formula 3 can be re-written with $Y = \alpha + x_n * \beta_n + \epsilon$, where "n" can be one or multiple predictors.

P7 (line 45): Why are the simple LMs necessary, if the effects of each weather indicator on mortality can also be quantified together in an MLR? A clear explanation or table of what models are calibrated to which datasets (e.g., national vs district level, with SLR and MLR applied), the number of parameters in each model, and the sample size of each dataset (including the districts) would be very useful to understand the overall statistical analysis done here.

P20: Table 1 is somewhat unclear. For example in the 1st row: 73% of 80 districts showed expected correlation, but of those only 15 had significant models? Is this interpretation correct? Furthermore, it seems as if only models that match the assumed correlation are included. It may be helpful to move "No of districts matching assumed correlation" to the column to the right of "No. of districts with valid data", and then next have a column with "No. of districts with significant models matching assumed correlation" and then a final column with "No. of districts with significant models not matching assumed correlation".

Table 1, Figure 3, Figure 4: The R^2 values reported here are concerning. Table 1 shows that the quality of fit of the linear models is quite weak ($R^2 < 0.10$) at the country domain, and the R^2 would be expected to decline even further when using the LMs to predict on independent data points. The higher median R^2 values at the district-level are slightly better, but the R^2 values in some districts (e.g., see max R^2 values in Figure 4) are very high and indicative of overfitting (i.e., model complexity / number of parameters approaches the sample size). This makes sense given the reportedly small sample sizes in the districts, but it raises two questions:

- 1) for models with a very low quality of fit, why bother with reporting or further interpretation of coefficients (as in Figure 4)?
- 2) where possible, why not quantify the predictive performance of these models by doing cross-validation to check for overfitting?

The authors stress in their discussion that the data meets the assumptions of the linear model, but do not address that the results show evidence of overfitting. Fitting simpler models with fewer parameters is one option, but with so few data points in some districts our ability to learn from the data is severely restricted. This is a limitation worth noting.

Have the authors analyzed the impact of Varroa mite control on the mortality rates? If so, it would be valuable to include this information to help readers understand how weather versus disease management influence mortality outcomes

===PREPARING YOUR MANUSCRIPT===

===PREPARING YOUR REVISION IN SCHOLARONE===

- If you are providing image files for potential cover images, please upload these at this step, and inform the editorial office you have done so. You must hold the copyright to any image provided.
- A copy of your point-by-point response to referees and Editors. This will expedite the preparation of your proof.

- Ensure that your data access statement meets the requirements at <https://royalsociety.org/journals/authors/author-guidelines/#data>. You should ensure that you cite the dataset in your reference list. If you have deposited data etc in the Dryad repository, please include both the 'For publication' link and 'For review' link at this stage.
- If you are requesting an article processing charge waiver, you must select the relevant waiver option (if requesting a discretionary waiver, the form should have been uploaded at Step 3 'File upload' above).
- If you have uploaded ESM files, please ensure you follow the guidance at <https://royalsociety.org/journals/authors/author-guidelines/#supplementary-material> to include a suitable title and informative caption. An example of appropriate titling and captioning may be found at https://figshare.com/articles/Table_S2_from_Is_there_a_trade-off_between_peak_performance_and_performance_breadth_across_temperatures_for_aerobic_scooping_in_teleost_fishes_/3843624.

Author's Response to Decision Letter for (RSOS-210618.R0)

See Appendix A.

RSOS-210618.R1 (Revision)

Review form: Reviewer 1

Is the manuscript scientifically sound in its present form?

Yes

Are the interpretations and conclusions justified by the results?

Yes

Is the language acceptable?

Yes

Do you have any ethical concerns with this paper?

No

Have you any concerns about statistical analyses in this paper?

No

Recommendation?

Accept as is

Comments to the Author(s)

The authors considered comments made by both reviewers and have rewritten the manuscript thoroughly. I'm OK with the current version of the manuscript.

I thank the Authors for their thorough review of the manuscript. It was a pleasure to review this paper.

Decision letter (RSOS-210618.R1)

Dear Mr Becsi,

It is a pleasure to accept your manuscript entitled "A biophysical approach to assess weather impacts on honey bee colony winter mortality" in its current form for publication in Royal Society Open Science. The comments of the reviewer(s) who reviewed your manuscript are included at the foot of this letter.

Kind regards,
Royal Society Open Science Editorial Office
Royal Society Open Science

on behalf of Dr Punidan Jeyasingh (Associate Editor) and Pete Smith (Subject Editor)
openscience@royalsociety.org

Associate Editor Comments to Author (Dr Punidan Jeyasingh):

Associate Editor: 1

Comments to the Author:

With much gratitude to the expert reviewers, I am delighted to recommend this manuscript for publication. Best wishes to the authors.

Reviewer comments to Author:

Reviewer: 1

Comments to the Author(s)

The authors considered comments made by both reviewers and have rewritten the manuscript thoroughly. I'm OK with the current version of the manuscript.

I thank the Authors for their thorough review of the manuscript. It was a pleasure to review this paper.

Appendix A

To the associate editor Dr Punidan Jeyasingh and two anonymous reviewers,

Thank you very much for the fair review process and the very helpful comments. We feel that these comments greatly improved the quality of the manuscript.

Below you find a point-by-point reply to the comments from the Associate Editor and the reviewers.

Benedikt Becsi on behalf of all authors.

Associate Editor Comments to Author (Dr Punidan Jeyasingh)

This manuscript takes a novel approach toward understanding the links between weather and honeybee mortality. The manuscript was assessed by two experts. Both experts were enthusiastic of the work. Nevertheless, they have raised several key issues that require attention. I felt the reviews were fair, clear, and constructive. With much gratitude to the expert reviewers, I invite the authors to incorporate these comments and resubmit a fresh version.

Thank you very much for your kind words, and we hope that the resubmitted version of the manuscript meets the expectations of your journal.

Reviewer comments to Author

Reviewer 1

Comments to the Author(s)

Research Article „A biophysical approach to assess weather impacts on honey bee colony winter mortality in Austria” written by Benedikt Becsi, Herbert Formayer, and Robert Brodschneider.

The study focus on colony losses of managed Western honey bee, linking weather conditions throughout the year with Western honey bee colony winter mortality. The approach presented by the Authors is novel and interesting, and the study provide new, important data related to ecology and management of Western honey bee. I congratulate to the Authors for the idea of the study and all the work done. However, in my opinion, the results of the study are misinterpreted and the main, important, finding – that weather conditions, although might contribute to the bee mortality, were among less important factors – remains unseen during discussion of the results, in conclusion, and in an abstract. In my opinion the study, apart from discussion and conclusions, is substantively correct and interesting, therefore I suggest major revision of the current version of the manuscript, mainly related to critical and factual interpretation of the results of the study. Below I provide detailed comments.

Many thanks for the constructive criticism, we hope that we adjusted the manuscript according to your valuable comments. We have rewritten the discussion and conclusion chapters with the intention to stress that our findings demonstrate that weather conditions are only one (rather weak) factor contributing to colony losses. Other major factors, like varroa mite, hive management etc., have already been discussed in the original submission, but were more highlighted in this revision. Below we provide detailed replies to your comments.

Title: why it is important to emphasize Austria in the title? In my opinion the study provides important general knowledge. The current form of the title suggests only local importance.

Thank you for this comment. We understand this comment (and similar comments below) on “local importance”, and removed “in Austria” in the title and elsewhere. We believe that the findings are not only of local importance, but tried to point out that the results derive from data collected in Austria, which could make them applicable for similar bees under similar climatic conditions.

Page 3, lines 25 and 43 – I suggest to not use phrase “the honey bee” when relating to *Apis mellifera*, since worldwide also other species of honey bees exist.

On both occasions, we inserted “Western” before honey bee to be more precise.

I also suggest to not state in the manuscript that *Apis mellifera* is the most important insect kept by humans without providing appropriate arguments (with citations), rationalizing this statement. Since discussion related to importance of *A. mellifera* is not the aim of this study, the Authors might consider removing this sentence.

The sentence has been modified to reflect the economic importance, but not to sound like the honey bee is the most important insect.

Page 4, lines 4-8. Please explain why it is important or emphasize that cited studies were done in Austria and in Luxembourg. Do these studies provide important data and general knowledge about *A. mellifera* ecology and management or they provide only locally important data? If so, why are they important?

See above comment on local importance. At one occasion, we modified the sentence to retain the information that beekeeper survey data from Austria was used.

Page 4, lines 17-19. Why the main goal of the study was to do the study in Austria? Is it important?

The locality was removed in this sentence.

Page 4, lines 44-45. It is oversimplification that bees defecate outside of the hive to avoid the spread of pathogens. Please avoid such statements. Instead, it is better to write that such behaviour may be related to reduction of pathogen dispersion and therefore is evolutionary and ecologically relevant.

The sentence has been rewritten to focus on reduction of pathogen dispersion.

Page 5, lines 53-55. I propose to use something like “percentage of colonies lost per district” instead of “ratios of (total number of lost colonies per district /total number of wintered colonies per district) × 100”, since the latter is hard to understand.

The sentence has been rewritten as suggested.

Discussion and Conclusion. In my opinion the whole discussion and conclusions should be rewritten, since in the current version of the manuscript the results are misinterpreted. While discussing the results the Authors focus very much on the effects of studied weather conditions on the bee mortality, suggesting that these effects are strong, undoubtful, and that the results of the study prove that the studied weather conditions are important factor driving Western honey bee colony winter mortality. However, the results suggest the opposite. Although the effects of studied weather conditions on the bee mortality were statistically significant, they were also weak. Therefore, the explanatory power of the weather conditions on the bee mortality is low. This means that not studied factors, other than weather conditions, shaped the bee mortality and that weather conditions are among less important factors. This is important result of the study, and it should be emphasized. I suggest focussing on this result during rewriting of the manuscript.

We recognise that these sections did not clearly reflect the model results, which suggest rather low explained variance of colony winter mortality by weather conditions. Many paragraphs discussing the model results and the Conclusions chapter were rewritten with a focus on quantifying the explanatory power of weather conditions, and what other factors might contribute to the large unexplained variance. Little is still known about the compared

relevance of explanatory factors from the literature, and how they in turn could be weather-dependent (indirect weather effects, e.g. on *Varroa* infestation).

Page 13, lines 21-25. The Authors suggest here that beekeepers may directly apply the results of this study. I do not agree with this statement, because in my opinion professional scientific knowledge, related among others to modelling, advanced statistics and scientific jargon, is needed for appropriate understanding of this manuscript.

We agree with the reviewer. We wrote this sentence to stress the applied importance of our findings, but as we received the feedback of the reviewer, we decided to remove this section.

Concluding section of the manuscript is, in my opinion, misleading. It should clearly state that studied factors had low explanatory power.

Conclusion chapter was rewritten with a clearer depiction of the relevance of studied factors (see above).

Reviewer 2

Comments to the Author(s)

Becsi et al. present an interesting study proposing hypotheses about how climate and weather conditions affect bee hive survival/mortality in apiaries across Austria. The authors are to be commended for clearly formulating and presenting their hypotheses based on mechanistic biophysical processes, and for deriving weather indicators that reflect their hypotheses. Moreover, the datasets used in this study are extensive in both their spatial extent (across a range of environmental conditions) and sample size. Given the complexity of these data sets and biological processes, the authors have developed an effective strategy for analyzing these data and testing their hypotheses. However, the manuscript would benefit greatly from (i) clarifications of the methods, (ii) explanations for why alternative methods were not used (in specific instances), and (iii) addressing limitations of model performance.

Thank you very much for your tremendously helpful and detailed comments. Your methodological suggestions really helped us address some overlooked issues and we feel that the manuscript improved substantially because of them. We clarified some ambiguous passages and provided additional tables to better explain our methodology. We briefly introduce alternative methods and provide arguments for the ones we have used. We have also rewritten portions of the results, discussion and conclusions chapters to reflect more on the limitations of our model results.

Below we provide detailed, point-by-point replies to your comments.

P3: The abstract does not report any quantitative results (e.g., quality of fit of the models). The authors should add information in the abstract which makes it clear that the models do not explain a high amount of the variation in mortality, so that readers who do not review the entire manuscript in detail are aware of the limitations.

Thanks, the abstract was amended with specifications about model performance.

P4 (lines 4-8): The authors cite Beyer et al (2018), Van Esch (2020), Calovi et al (2021) later in the manuscript, but should reference these studies here, as examples of empirical studies evaluating the role of meteorological conditions across the year in influencing winter mortality.

The reviewer is correct, we amended accordingly, though the Beyer et al. (2018) article was already cited in this paragraph in the original submission.

P4 (lines 52-53): The authors should add text explaining that the bees must move around to get to honey stores, but cannot break cluster if temps are too low

Thanks, information on this was added.

P4 (line 54): "at this time, food demand is increased: due to the start of breeding" maybe rephrase to "start of egg-laying"?

We included both, the sentence now reads: “At this time, food demand is increased due to the start of egg-laying and brood rearing.”

P5 (lines 53-60): The authors justify the mortality z-score because it enables relative comparisons of mortality rates among the districts, and this seems a valid reason. However, why not use binary observations of bee hive mortality and apply a logistic regression (i.e., a generalized linear model) or tree-based machine learning algorithm?

Thank you for the suggestion. Using a binary target variable is not supported by the beekeeper survey data, because it only contains data on beekeeper level, not on hive level. So the smallest data unit of the survey is the number of lost colonies and the total number of colonies per beekeeper, which has been aggregated to district level to reduce inaccuracies in the individual survey responses.

Using a regression tree is possible, but would require additional constraints for the model to avoid overfitting, and is generally more complex to interpret than manually selecting the input variables/geographical subsets, which helped verify our hypotheses. Therefore, we decided to keep the original methodological approach with single and multiple linear models, but amended the methods section with a note that tree-based machine learning can also be applied to the problem.

P5 (lines 46-47): The study aggregates the respondents' point locations to polygons representing the 94 political districts in Austria. This is an example of the modifiable areal unit problem, which can induce statistical bias due to the pattern of spatial aggregation (according to political boundaries, no less!). Have the authors considered how alternative methods of aggregating the points may result in biased observations?

Indeed the aggregation to political boundaries is problematic. The chosen unit of aggregation stems from the original study on the correlations between bee colony winter mortality and weather conditions, which the authors carried out for the Federal State of Upper Austria. The region of interest of this study was the administrative boundaries of public authorities. We introduced several constraints to the aggregation to avoid bias, but realise that other areal units (e.g. hexagons) could reduce this even further. We addressed this issue in the first paragraph of the 'Mortality rates' chapter.

P6 (lines 1-7): So M_z is the z-score of the mortality rate in a district in a given year? Perhaps the equation should include indices and more detailed definitions to more clearly present what is being calculated

Yes, that's correct. The formula and definitions were complemented accordingly.

P6 (lines 21-51): The authors do not provide information about the correlation structure of dependent weather variables, which would complicate parameter estimation

We added a new table to the supplementary material (Tab. S.1 (new)) that provides the correlations between the independent variables. It is referenced in the 'Weather indicators' and 'Regression analysis' sections in the 'Methods' chapter.

P6 (line 60): Do the authors mean to say that temperature-dependent indicators are highly correlated with elevation, not sea-level?

Thank you for noticing!

P7 (line 5): For some of the districts, these masked areas seem to cover most of the district – is this a problem when evaluating how meteorological conditions influence outcomes in these districts?

Not really, since also the mean meteorological conditions of a district are only calculated from data of areas below 1.200m. Even though large portions of some districts feature higher elevation, the vast majority of beekeepers operate in the lower elevations.

P7 (line 14): Again, here would be a good place to mention any strong correlations among the independent variables.

Done!

P7 (line 25): The definition of the $x \cdot \beta$ term in Formula 2 should be: β is the regression coefficient/parameter and x are the model inputs.

Thank you for the suggestion, the definition was added to the formula.

P7 (line 20-44): It may be simpler to combine Formula 2 and Formula 3 to define the simple linear regression (SLR) and multiple linear regression (MLR) models with one equation. Formula 3 can be re-written with $Y = \alpha + x_n \cdot \beta_n + \epsilon$, where “n” can be one or multiple predictors.

Formulas 2 and 3 were combined as recommended.

P7 (line 45): Why are the simple LMs necessary, if the effects of each weather indicator on mortality can also be quantified together in an MLR? A clear explanation or table of what models are calibrated to which datasets (e.g., national vs district level, with SLR and MLR applied), the number of parameters in each model, and the sample size of each dataset (including the districts) would be very useful to understand the overall statistical analysis done here.

Our line of thought here was that for testing our four hypotheses, it would be useful to not only present the significance of each indicator in a MLR, but also quantify the predictive performance of a model that includes only the single indicator underpinning a hypothesis. It was interesting to learn how the weather indicators performed in comparison to each other, and what geographical patterns were unveiled by the simple LMs on district level. We appreciate the suggestion of a table that provides a clear overview of the statistical analyses and have added it to the supplementary (Tab. S.2) and referenced it in the final paragraph of the ‘Methods’ section.

P20: Table 1 is somewhat unclear. For example in the 1st row: 73% of 80 districts showed expected correlation, but of those only 15 had significant models? Is this interpretation

correct? Furthermore, it seems as if only models that match the assumed correlation are included. It may be helpful to move “No of districts matching assumed correlation” to the column to the right of “No. of districts with valid data”, and then next have a column with “No. of districts with significant models matching assumed correlation” and then a final column with “No. of districts with significant models not matching assumed correlation”.

Thank you for the suggestion to clarify, table 1 has been adapted accordingly.

Table 1, Figure 3, Figure 4: The R^2 values reported here are concerning. Table 1 shows that the quality of fit of the linear models is quite weak ($R^2 < 0.10$) at the country domain, and the R-squared would be expected to decline even further when using the LMs to predict on independent data points. The higher median R^2 values at the district-level are slightly better, but the R^2 values in some districts (e.g., see max R^2 values in Figure 4) are very high and indicative of overfitting (i.e., model complexity / number of parameters approaches the sample size). This makes sense given the reportedly small sample sizes in the districts, but it raises two questions:

- 1) for models with a very low quality of fit, why bother with reporting or further interpretation of coefficients (as in Figure 4)?
- 2) where possible, why not quantify the predictive performance of these models by doing cross-validation to check for overfitting?

Yes, the country-wide models suffer from a lot of noise (short time series over a lot of different categories/districts), and the multivariate model on the country domain could not explain more than 10% of the variance. For us it was important to see if the regression slopes would match our expected correlations from the hypothesis, which they did. On district level, we only considered the models with the expected slope, therefore the median R^2 -values are higher here. The point about quantifying overfitting is a good suggestion. We performed leave-one-out cross validation for all multivariate district models and found out that the median R^2 for all valid districts is about 0.2 (compared to ~ 0.4 for the models including all observations). This means that when corrected for overfitting, the multivariate models could not produce a higher quality of fit than the best performing single indicator models with expected regression slope. This point was included in the paragraphs of the discussion chapter that reflect on the regression model results. The results of the cross validation were added to Tab.S.3 and also referenced in the final paragraph of the results chapter.

The authors stress in their discussion that the data meets the assumptions of the linear model, but do not address that the results show evidence of overfitting. Fitting simpler models with fewer parameters is one option, but with so few data points in some districts our ability to learn from the data is severely restricted. This is a limitation worth noting.

The reviewer is correct. Additional text passages noting the limitations of the multiple predictor models on district scale due to overfitting were added to the paragraphs P12 (line 13-23) and P12 (line 25-39).

Have the authors analyzed the impact of Varroa mite control on the mortality rates? If so, it would be valuable to include this information to help readers understand how weather versus disease management influence mortality outcomes

The varroa mite treatments are available from the COLOSS questionnaire with a monthly solution. The methods applied are rather homogeneous in Austria (mostly formic acid treatments in Summer and oxalic acid treatments in winter). For the very interesting question raised by the reviewer, higher temporal resolution, more details on the application of acids (evaporator types, etc.) and probably microclimatic data would be required, which is not available.